# GUI-ReWalk: Massive Data Generation for GUI Agent via Stochastic Exploration and Intent-Aware Reasoning

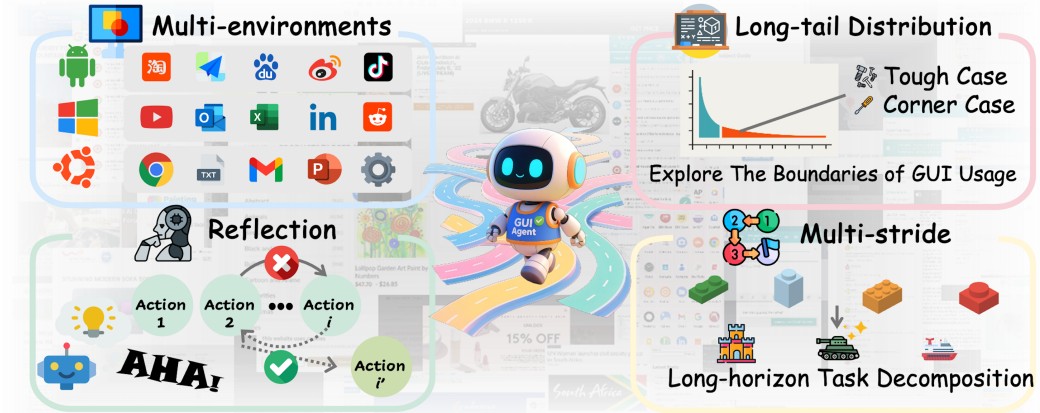

Figure 1: Illustration of GUI-ReWalk Characteristics – Multi-Platform Coverage, Long-Tail Patterns, Reflective Learning, and Multi-Stride Workflows.

## Abstract

Graphical User Interface (GUI) Agents, powered by large language and vision-language models, hold promise for enabling end-to-end automation in digital environments. However, their progress is fundamentally constrained by the scarcity of scalable, high-quality trajectory data. Existing data collection strategies either rely on costly and inconsistent manual annotations or on synthetic generation methods that trade off between diversity and meaningful task coverage. To bridge this gap, we present **GUI-ReWalk**—a reasoning-enhanced, multi-stage framework for synthesizing realistic and diverse GUI trajectories. GUI-ReWalk begins with a stochastic exploration phase that emulates human trial-and-error behaviors, and progressively transitions into a reasoning-guided phase where inferred goals drive coherent and purposeful interactions. Moreover, it supports multi-stride task generation, enabling the construction of long-horizon workflows across multiple applications. By combining randomness for diversity with goal-aware reasoning for structure, GUI-ReWalk produces data that better reflects the intent-aware, adaptive nature of human-computer interaction. We further train GUI-ReWalk-7B on the our dataset and evaluate it across multiple benchmarks, including Screenspot-Pro, OSWorld-G, UI-Vision, AndroidControl, and GUI-Odyssey. Results demonstrate that GUI-ReWalk enables superior coverage of diverse interaction flows, higher trajectory entropy, and more realistic user intent. These findings establish GUI-ReWalk as a scalable and data-efficient framework for advancing GUI agent research and enabling robust real-world automation.

# 1 INTRODUCTION

The emergence of Vision-Language Models (VLMs) has significantly advanced the capabilities of autonomous agents in perceiving, reasoning, and acting within complex environments (Zhang et al., 2025b; Wang et al., 2025). A promising and increasingly popular research direction is that of GUI Agents, where large language model (LLM)-based agents interact with Graphical user interfaces (GUIs) to accomplish real-world tasks. By bridging visual perception, semantic understanding, and action planning, GUI Agents are poised to unlock a new era of end-to-end automation, transforming how intelligent systems interact with the digital world across domains ranging from productivity to everyday services.

However, the development of GUI Agents is currently constrained by the availability of high-quality training data. Existing GUI agent trajectories are primarily obtained through manual annotation or synthetic generation. Manual collection involves labeling complete action trajectories and defining high-level tasks, a process that is not only time-consuming and labor-intensive, but also susceptible to inconsistencies in quality and style due to varying annotator expertise. On the other hand, synthetic data generation is typically driven by either predefined task goals or random environment interaction. Task-driven approaches offer clear and structured objectives, but suffer from limited scalability and diversity. In contrast, interaction-based methods promote trajectory diversity, yet often lead to overly divergent behaviors that fail to converge on meaningful task outcomes.

Unlike traditional text or image data, GUI trajectories embody rich patterns of human interaction with graphical interfaces. They are not simple Markovian sequences, but rather unfolding narratives shaped by both intention and exploration. Human behavior in GUI environments typically unfolds through the following progressive stages:

- Exploration and boundary probing: When first encountering an unfamiliar application or interface, users often engage in seemingly random tapping, swiping, and navigating actions to test affordances and interface boundaries;

- Goal formulation and pursuit: As users develop clearer objectives, their actions become more directed and intentional, focusing on accomplishing specific tasks through iterative interactions;

- Cross-application coordination: To fulfill more complex goals, users frequently orchestrate multiple apps in tandem;

- Self-correction and backtracking: Users identify missteps or unreachable states and adapt by revising their strategies, undoing actions, or restarting from known checkpoints.

These patterns reveal that GUI trajectories are neither purely random nor rigidly deterministic—they embody a delicate balance between "chaos" and "order," being structured, goal-driven, and highly adaptive.

To address the limitations of existing data acquisition approaches and better capture the nuanced characteristics of human GUI behavior, we propose **G**raphical **U**ser **I**nterface **Re**asoning and random **Walk** (**GUI-ReWalk**)—a multi-stage framework that integrates stochastic exploration with goal-directed reasoning to synthesize diverse and realistic GUI trajectories. Inspired by how humans explore unfamiliar interfaces, GUI-ReWalk begins with a random walk phase, simulating natural trial-and-error behaviors akin to an uninformed policy over a Markov chain, where each state transition depends only on the current state and available actions. As the trajectory unfolds, a large language model (LLM) acts as a reasoning agent that interprets the partially observed sequence and infers high-level goals, transitioning the framework into a reasoning-guided phase. This phase resembles a policy update in a hierarchical Markov Decision Process (hMDP), where action generation is conditioned not only on the current GUI state but also on the inferred intent—mirroring how users refine their behavior upon gaining contextual understanding. In addition, GUI-ReWalk supports multi-stride task generation, where each stride represents a subtask composed of several low-level actions, sequentially coordinated to complete complex goals across multiple interfaces or applications. By unifying the randomness of Markov chains with the intent-aware adaptability of hMDPs, GUI-ReWalk produces synthetic interaction data that captures both the long-tail diversity and the structured, goal-driven nature of real-world human-computer interaction.

In our experiments, we developed GUI-ReWalk-7B, built on Qwen2.5-VL-7B, and trained it on synthetic trajectory data generated within a controlled GUI environment. We evaluated its grounding and navigation capabilities across multiple benchmarks, including Screenspot-Pro, OSWorld-G, and UI-Vision for grounding, and AndroidControl and GUI-Odyssey for navigation. Results demonstrate that GUI-ReWalk, leveraging systematic trajectory generation and task-aware supervision, achieves superior coverage of diverse interaction flows, higher trajectory entropy, and realistic user intent, as validated by human evaluations. These findings establish GUI-ReWalk as a highly effective, scalable, and data-efficient solution for advancing human-computer interaction in diverse GUI environments.

In summary, our work makes the following key contributions:

- **Human-like modeling of GUI interaction:** We formalize GUI trajectories as a hierarchical Markov Decision Process, where each stride combines subgoal abstraction with stride-based reasoning to capture both exploratory and goal-directed behaviors.

- **The GUI-ReWalk framework:** We introduce a multi-stage framework integrating random exploration, task-guided completion, and cross-application task initiation, enhanced by retrospective LLM-based annotation and error-recovery mechanisms that mirror real human interaction patterns.

- **Dataset analysis and model evaluation:** We provide an in-depth analysis of the GUI-ReWalk dataset and demonstrate its effectiveness by training GUI-ReWalk-7B, which achieves substantial improvements across grounding and navigation benchmarks.

## 2 RELATED WORKS

### 2.1 EVOLUTION OF GUI AGENTS

GUI agents have progressively evolved from rule-based systems to data-driven, end-to-end models. Early approaches—including RPA tools (Dobrica, 2022; Hofmann et al., 2020), DART (Memon et al., 2003), and World of Bits (WoB) (Shi et al., 2017)—relied on predefined heuristics and API invocations to mimic user actions, but exhibited limited flexibility and poor generalization in dynamic or unfamiliar environments. The emergence of modular agent frameworks—integrating foundation models (e.g., GPT-4o (OpenAI, 2024)), memory systems (e.g., Cradle (Tan et al., 2024)), grounding components (e.g., MM-Navigator (Yan et al., 2023)), and tool-use mechanisms (e.g., AutoGPT (Yang et al., 2023))—enabled more adaptive and multi-step interactions. However, these systems often remained constrained by handcrafted workflows, prompt engineering, and brittle module coordination (Xia et al., 2024).

More recently, native agent architectures such as Claude Computer Use (Anthropic, 2024), Aguvis (Xu et al., 2025b), OS-Atlas (Wu et al., 2024), and UI-TARS (Qin et al., 2025) have unified perception, reasoning, memory, and action within end-to-end, vision-centric models. These agents operate directly on raw screenshots without relying on structured UI representations (e.g., accessibility trees or HTML), and are trained on large-scale GUI interaction data, achieving improved generalization across platforms such as web, mobile, and desktop. Building on this foundation, recent work has further enhanced native agents through targeted training strategies—including reinforcement learning, supervised fine-tuning, and curriculum learning—along with dedicated datasets for grounded interaction (Yang et al., 2025; Wu et al., 2025b; Tang et al., 2025a; Park et al., 2025; Lian et al., 2025; Tao et al., 2025; Chen et al., 2025b), complex task reasoning (Tang et al., 2025b; Lu et al., 2025; Wei et al., 2025; Xie et al., 2025c), and reflective decision-making (Wu et al., 2025a; Wanyan et al., 2025).

### 2.2 GUI BENCHMARKS AND ENVIRONMENTS

Benchmark environments play a central role in the development of GUI agents by defining interaction modalities, task formats, and evaluation protocols. Early benchmarks such as Mini-Wob++ (Liu et al., 2018) and WoB (Shi et al., 2017) provided synthetic but controlled environments—MiniWob++ emphasized UI layout and instruction diversity, while WoB enabled reproducible task execution on real webpages. Subsequent benchmarks moved toward greater realism

and task complexity. WebShop (Yao et al., 2022) introduced compositional shopping tasks requiring semantic reasoning and goal-driven navigation, and Mind2Web (Deng et al., 2023) scaled to 2,000 open-ended tasks across 137 websites with fine-grained step annotations. WebArena (Zhou et al., 2024) and VisualWebArena (Koh et al., 2024) simulated multimodal websites across diverse domains (e.g., e-commerce, social media), while WebLINX (Lù et al., 2024) extended to long-horizon, multi-turn workflows using retrieval-augmented prompting and expert demonstrations.

Beyond the browser, benchmarks such as OSWorld (Xie et al., 2024) and WindowsAgentArena (Bonatti et al., 2024) enabled agents to interact with full desktop operating systems, supporting complex workflows like file management and multi-application coordination. On mobile platforms, AndroidWorld (Rawles et al., 2025) and GUI-Odyssey (Lu et al., 2024) enabled fine-grained UI interactions across and within apps. Finally, modality-rich and cross-platform benchmarks have emerged to support generalist agents: GUI-World (Chen et al., 2025a) captured video-based GUI behavior grounded in real-world demonstrations, while AgentSynth (Xie et al., 2025a) introduced a modular benchmark that generates long-horizon desktop tasks from atomic subtasks via LLMs, facilitating structured evaluation of planning, perception, and robustness.

## 2.3 Data Collection and Synthesis for GUI Agents

Training GUI agents depends on large-scale, diverse task trajectories. Early datasets such as Web-Shop (Yao et al., 2022), Mind2Web (Deng et al., 2023), and AndroidControl (Li et al., 2024) were constructed through human demonstrations to ensure task fidelity and realism. GUI-Odyssey (Lu et al., 2024) further contributed 7,700 mobile interaction episodes spanning both within-app and cross-app workflows. However, the scalability of these human-annotated datasets is hindered by high collection costs.

To overcome this limitation, recent efforts have explored automated data generation techniques. OS-Genesis (Sun et al., 2025) extracts high-quality task trajectories via agent-driven exploration guided by learned reward models. WebSynthesis (Gao et al., 2025) performs world-model-guided search over simulated web interfaces to synthesize interaction traces. GUI-World (Chen et al., 2025a) generates video-based interaction data from curated app screenshots, while TongUI (Zhang et al., 2025a) mines web tutorials and converts them into over 143K multimodal, executable task trajectories grounded in realistic application scenarios.

## 3 GUI-ReWalk

### 3.1 Overview

The GUI-ReWalk framework is designed to replicate the iterative, exploratory, and goal-oriented behaviors characteristic of human interactions with GUIs. To this end, we formalize the framework as hierarchical Markov Decision Process, consisting of multiple sequentially executed steps. Each stride consists of three distinct phases: a random walk phase, a task-guided completion phase, and a task initiation phase in cross-application. The GUI-ReWalk framework integrates both random exploratory actions and logical reasoning processes, thereby enhancing the diversity and length of interaction trajectories. Between these phases, we introduce retrospective annotation, which uses large language models (LLMs) to perform backward annotation and summarization of trajectories, enabling full automation of the process. In both the task-guided completion phase and the task initiation phase, we further incorporate a error task recovery scheme that generates a new goal whenever the LLM becomes blocked in the current environment, drawing upon the previously failed objective and the current state to ensure continuity and robustness in task execution.

Through this mechanism, GUI-ReWalk effectively generates multi-stride trajectories that closely mirror human multi-application workflows, preserving logical task coherence while fluidly transitioning between exploratory and goal-directed behaviors. When the framework encounters a dead end that prevents task completion, it invokes a reflective reasoning process to revise the original objectives, leveraging both the initial goals and the interaction history, to formulate new, relevant, and executable targets. This capability enables the system to recover progress, thereby ensuring the continuity and robustness of task execution.

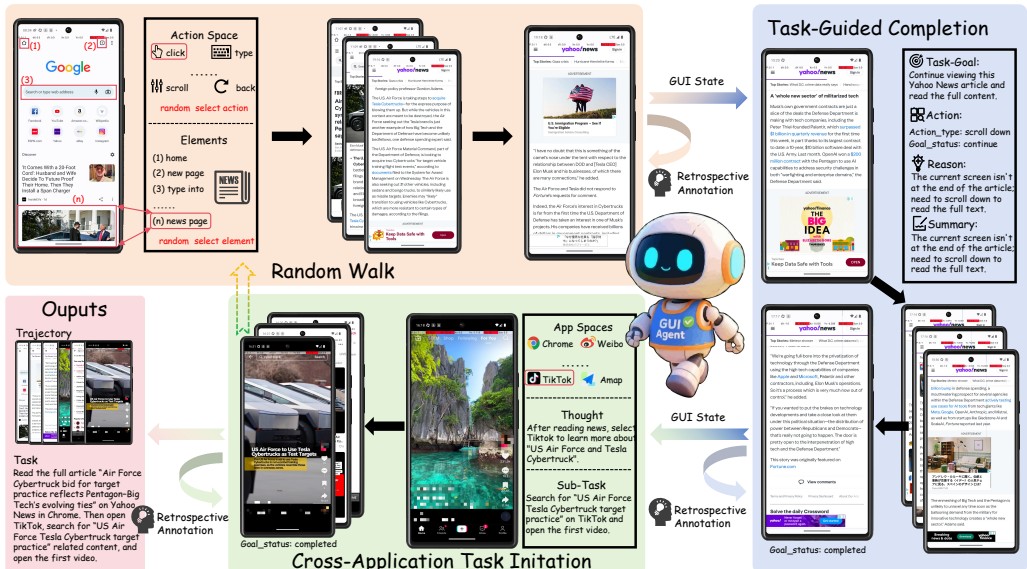

Figure 2: Overview of GUI-ReWalk Framework. Starting from a random app, GUI-ReWalk performs **Random Walk** by selecting actions and interacting with elements step by step; it then transitions to **Task-Guided Completion** to complete minimal-step tasks forming a stride, followed by **Cross-Application Task Initiation** to propose and execute new tasks in related apps. After each sub-stage, **Retrospective Annotation** records executed actions and GUI states. This cycle repeats across multiple strides to generate complete trajectories and overall task objectives.

## 3.2 RANDOM WALK

The random walk phase reflects human exploration and boundary probing when interacting with unfamiliar interfaces. In this phase, users often engage in trial-and-error actions without a clear objective. GUI-ReWalk models this behavior as a Markov chain:

$$\mathcal{M}_r = (\mathcal{S}, \mathcal{A}^r, P^r),$$

where $\mathcal{S}$ denotes the state of the GUI environment, and $\mathcal{A}^r$ denotes the primitive GUI actions available in this state. The state transition probability of the random walk is defined as:

$$P^r(s_{t+1} \mid s_t) = \sum_{a_t^r} P^r(s_{t+1} \mid s_t, a_t^r) \, P^r(a_t^r \mid s_t),$$

where $s_t \in \mathcal{S}$ and $a_t^r \in \mathcal{A}^r$ denote the state and accessible actions at time step $t$. During the exploration phase, the action policy is set to a uniform distribution $P^r(a_t^r \mid s_t) = \frac{1}{|\mathcal{A}^r|}$ to maximize state-space coverage and emulate the chaotic probing behavior commonly observed in human users.

After this, GUI-ReWalk randomly chooses an executable UI element for the selected action. For input-type actions, such as typing, GUI-ReWalk leverages the LLM to generate context-appropriate text. As the trajectory stride extends over multiple iterations, the length of the random walk is gradually reduced to better reflect the natural shift from broad exploration to focused interaction observed in real human behavior.

## 3.3 TASK-GUIDED COMPLETION

After exploring the environment, human users typically form explicit goals and act purposefully. GUI-ReWalk models the task-guided completion phase as a goal-constrained Markov decision pro-

cess (Kaelbling et al., 1998). In this process, we first use the LLM to infer a high-level task goal from the terminal exploration state.

$$g = \Phi_{\text{LLM}}(s_{T_r}), \quad s_{T_r} \in \mathcal{S},$$

where $\Phi_{\text{LLM}}$ is the LLM goal inference function, $T_r$ is the terminal time step of the random walk. Then the task-guided completion phase is formalized as:

$$\mathcal{M}_g = (\mathcal{S}, \mathcal{A}^g, P^g, \mathcal{R}^g, \pi),$$

The state transition probability of task-guided completion captures how humans act purposefully once they have a clear goal in mind. It can be formulated as:

$$P^t(s_{t+1} \mid s_t) = \sum_{a_t^g} P^t(s_{t+1} \mid s_t, a_t^g) \, \pi(a_t^g \mid s_t),$$

where $a_t^g \in \mathcal{A}^g$ and $\pi(a_t^g \mid s_t)$ is the action policy based on the LLM. $\pi(a_t^g \mid s_t)$ selects the next action based on the current state and the intended goal. When an action cannot be executed within the current environment, $\pi(a_t^g \mid s_t)$ engages a reflective reasoning process to revise the goal $g$, ensuring that the updated objective remains both relevant and feasible for continued task execution. To reflect the sparsity of meaningful task completion signals, we define a sparse reward (Andrychowicz et al., 2017; Schaul et al., 2015) function as:

$$\mathcal{R}^g = \begin{cases} r_{\text{succ}} = 1, & \text{if } s \in S^g, \\ 0, & \text{otherwise,} \end{cases}$$

where $\mathcal{S}^g = \mathcal{S} \times \mathcal{G}$ is the task-conditioned state space and $\mathcal{G}$ is the goal space. The prompts for $\Phi_{\text{LLM}}$ and $\pi(a_t^g \mid s_t)$ are shown in the E.1.

### 3.4 CROSS-APPLICATION TASK INITIATION

Similar to the goal inference in the task-guided completion phase, GUI-ReWalk leverages the LLM to analyze the trajectory and annotations of the current stride $E_i$. Based on this analysis, it generates a semantically related cross-application goal to initiate the next stride:

$$G_{i+1} = \Pi_{\text{LLM}}(s_i),$$

where $s_i$ denotes the final state of the $i$-th stride, and $\Pi_{\text{LLM}}$ is the goal-generation policy implemented by the LLM. The inferred goal $G_{i+1}$ is then used to initialize the next stride $\mathcal{L}_{i+1}$. Upon switching to a new application, GUI-ReWalk re-enters the process, performing a random walk, task-guided completion, and retrospective annotation, thereby constructing a new stride that continues the multi-application trajectory.

This hierarchical orchestration mirrors human multi-application workflows, where users frequently transition from completing one task to initiating another related task across different applications. It maintains the same alternation between chaotic exploration and goal-directed execution, while ensuring semantic continuity across strides to form coherent, multi-application trajectories. The prompt for $\Pi_{\text{LLM}}$ is shown in the E.2.

### 3.5 RETROSPECTIVE ANNOTATION

At the end of each phase, GUI-ReWalk performs retrospective annotation through the LLM. Retrospective Annotation serves as an automated alternative to manual labeling, enabling the generation of semantically rich supervision without human intervention:

$$\mathcal{B} : \tau = \{s_t, a_t\}_{t=1}^{T_g} \longmapsto \left(U_\tau, \{u_t\}_{t=1}^{T_g}\right),$$

Table 1: Performance comparison on grounding datasets. The reported scores represent the average performance across all sub-tasks within each benchmark.

| Model | Screenspot-Pro | OSWorld-G | UI-Vision |
|---|---|---|---|
| GPT-4o (OpenAI, 2024) | 0.8 | – | 1.4 |
| SeeClick-9.6B (Cheng et al., 2024) | 1.1 | – | 5.4 |
| OS-Atlas-7B (Wu et al., 2024) | 18.9 | 22.7 | 9.0 |
| UGround-7B (Gou et al., 2025) | 16.5 | 36.4 | 12.9 |
| UI-TARS-1.5-7B (Qin et al., 2025) | 46.4 | 45.5 | 20.3 |
| Qwen2.5-VL-7B (Bai et al., 2025) | 20.8 | 16.8 | 3.7 |
| GUI-ReWalk-7B (ours) | 35.1 | 27.5 | 5.9 |

Formally, given a transition triplet $\langle s_{t-1}, a_{t-1}, s_t \rangle \in \tau$, GUI-ReWalk employs an LLM to infer the corresponding step-level instruction $u_t$. This process yields a sequence $\{(s_t, u_t)\}_{t=1}^{T_g}$ that encapsulates fine-grained semantic guidance for each state. Subsequently, the full set of states and their associated step-level instructions are jointly considered to infer a high-level task description $U_\tau$ for the entire stride, thereby bridging low-level execution steps with the overarching task semantics.

## 3.6 TASK RECOVERY

Task recovery addresses scenarios where users deviate from their intended trajectory due to errors, ambiguous goals, or unforeseen interface dynamics. GUI-ReWalk models this recovery process as an adaptive replanning mechanism built on the interplay between state monitoring and LLM-driven reasoning.

Formally, when the agent detects that the current trajectory $\tau = \{s_t, a_t\}_{t=1}^T$ fails to progress toward the inferred goal $g$, a recovery trigger is activated:

$$\Omega(s_t, g) = \begin{cases} 1, & \text{if progress towards } g \text{ stalls or repeat}, \\ 0, & \text{otherwise}. \end{cases}$$

Once activated, we use the LLM to re-analyze the current environment to update or refine the task objective:

$$g' = \Psi_{\text{LLM}}(s_{t'}, g),$$

with $\Psi_{\text{LLM}}$ denoting the goal-revision function. This allows the agent to dynamically adapt its task representation when the original goal $g$ becomes infeasible or underspecified.

The subsequent execution continues under the revised policy $\pi'(a \mid s, g')$, ensuring that the trajectory realigns with a coherent objective. This recovery loop effectively captures human-like resilience in digital environments, enabling GUI-ReWalk to handle interruptions, erroneous actions, and semantic drift robustly. By incorporating task recovery, the framework closes the loop between exploration, goal-directed execution, and error correction, thereby achieving more reliable and human-aligned multi-application task automation.

## 4 EXPERIMENTS AND RESULTS

We train GUI-ReWalk-7B on trajectory data generated within our GUI environment, using Qwen-2.5VL-7B as the base model. The evaluation is conducted from two perspectives: grounding and navigation. For grounding, the full controllability of the GUI environment enables the construction of a large-scale dataset for model training. For navigation, we apply LLM-based automated filtering and trajectory scoring to select high-quality samples for supervised fine-tuning (SFT).

## 4.1 GROUNDING

We evaluate the grounding capability of GUI-ReWalk on several publicly available benchmarks, including Screenspot-Pro (Li et al., 2025), OSWorld-G (Xie et al., 2025b), and UI-Vision (Nayak

Table 2: Comparison of models on navigation benchmarks. "Type Acc." denotes type accuracy, and "SR" denotes success rate.

| Model | AndroidControl-Low | | AndroidControl-High | | GUI-Odyssey | |
|---|---|---|---|---|---|---|
| | Type Acc. | SR | Type Acc. | SR | Type Acc. | SR |
| GPT-4o (OpenAI, 2024) | 74.3 | 19.4 | 66.3 | 20.8 | 34.3 | 3.3 |
| SeeClick-9.6B (Cheng et al., 2024) | 93.0 | 75.0 | 82.9 | 59.1 | 71.0 | 53.9 |
| OS-Atlas-7B (Wu et al., 2024) | 93.6 | 85.2 | 85.2 | 71.2 | 84.5 | 62.0 |
| OS-Genesis-7B (Sun et al., 2025) | 90.7 | 74.2 | 66.2 | 44.5 | – | – |
| Qwen2.5-VL-7B (Bai et al., 2025) | 91.8 | 85.0 | 70.9 | 69.8 | 59.5 | 46.3 |
| GUI-ReWalk-7B (ours) | 91.7 | 96.3 | 73.1 | 66.2 | 69.6 | 64.2 |

et al., 2025). The overall results are reported in Table 1, with a detailed breakdown provided in the Appendix.

**ScreenSpot-Pro** targets professional software with high-resolution interfaces. It emphasizes grounding in visually complex environments, where dense and heterogeneous iconography poses substantial challenges. As shown in Table 1, with 100k generated grounding data, GUI-ReWalk-7B improves upon Qwen2.5-VL-7B by 14.3.

**OSWorld-G** consists of fine-grained tasks that closely simulate authentic computer usage. It provides a holistic evaluation of grounding in real-world digital environments. GUI-ReWalk-7B yields an improvement of 10.7 over Qwen2.5-VL-7B.

**UI-Vision** evaluates grounding performance across diverse and fine-grained tasks in realistic desktop environments, offering a comprehensive assessment of practical grounding capabilities. GUI-ReWalk-7B achieves an improvement of 2.2 compared with Qwen2.5-VL-7B.

In conclusion, GUI-ReWalk delivers improvements in grounding performance across professional software, realistic desktop tasks, and fine-grained computer-use scenarios. Compared with general-purpose vision–language models of the same scale (e.g., Qwen2.5-VL-7B), GUI-ReWalk demonstrates significantly stronger capabilities. These results establish GUI-ReWalk as a more reliable and generalizable solution among models of comparable size, highlighting its potential to advance grounding in practical human–computer interaction tasks.

## 4.2 NAVIGATION

To evaluate the multi-step decision-making capability of our proposed method, GUI-ReWalk, we conduct experiments on several publicly available benchmarks, including AndroidControl (Li et al., 2024) and GUI-Odyssey (Lu et al., 2024). The overall results are summarized in Table 2, with detailed analyses provided in the Appendix.

**AndroidControl** is a static offline benchmark designed to evaluate UI comprehension, task decomposition, and action planning under both low-level and high-level task instructions. Compared with Qwen2.5-VL-7B, GUI-ReWalk achieves consistent improvements across both evaluation metrics. Specifically, on low-level tasks, GUI-ReWalk maintains roughly flat type accuracy and boosts step success rate by 11.3. On high-level tasks, it further achieves gains of 2.2 in type accuracy. These results highlight the model's superior ability in hierarchical planning and abstraction.

**GUI-Odyssey** provides complementary offline evaluation tasks that emphasize structured reasoning and robust action planning in controlled environments. On this benchmark, GUI-ReWalk outperforms Qwen2.5-VL-7B by 10.1 in type accuracy and 17.9 in step success rate, further validating the model's effectiveness in offline multi-step decision-making and complex task decomposition.

To investigate the impact of trajectory stride on model performance, we conduct an ablation study by varying the stride number during training. Specifically, we evaluate GUI-ReWalk with 1, 2, and 3 strides across navigation benchmarks. The results are summarized in Table 3.

The results indicate that stride size plays a critical role in balancing training efficiency and performance. Using a single stride limits the diversity of trajectory supervision, leading to suboptimal

Table 3: Ablation study of trajectory stride number on GUI-ReWalk. "Type Acc." denotes type accuracy, and "Value Acc." denotes value accuracy.

| Stride Number | AndroidControl-Low | | AndroidControl-High | | GUI-Odyssey | |
|---|---|---|---|---|---|---|
| | Type Acc. | Value Acc. | Type Acc. | Value Acc. | Type Acc. | Value Acc. |
| Stride = 1 | 91.7 | 31.5 | 73.1 | 30.6 | 69.6 | 33.0 |
| Stride = 2 | 91.9 | 32.1 | 72.7 | 38.6 | 70.8 | 39.4 |
| Stride = 3 | 92.0 | 32.3 | 72.5 | 37.9 | 72.2 | 39.7 |

Table 4: Unified action space for different environments.

| Environments | Action | Definition | Mobile Rate | Desktop Rate |
|---|---|---|---|---|
| Shared | Click(x, y) | Clicks at coordinates (x, y). | 61.67% | 78.49% |
| | Scroll(direction) | Scrolls the screen with specified direction. | 9.31% | 1.03% |
| | Drag(x1,y1, x2,y2) | Drags from (x1, y1) to (x2, y2). | 0.05% | 1.22% |
| | Type(content) | Types the specified content. | 9.19% | 4.00% |
| | Wait() | Waits for screen update. | 3.14% | 0.91% |
| | Completed() | Marks the task as finished. | 7.79% | 5.62% |
| | Infeasible() | Marks the task as cannot be done. | 0.56% | 1.68% |
| Mobile | Launch(app) | Opens the specified app. | 7.53% | - |
| | LongPress(x, y) | Long presses at (x, y). | 0.21% | - |
| | PressBack() | Presses the "back" button. | 0.32% | - |
| | PressHome() | Presses the "home" button. | 0.16% | - |
| | PressEnter() | Presses the "enter" key. | 0.07% | - |
| Desktop | HotKey(key) | Performs the specified hotkey. | - | 0.59% |
| | LeftDouble(x, y) | Double-clicks at (x, y). | - | 4.33% |
| | RightSingle(x, y) | Right-clicks at (x, y). | - | 2.13% |

grounding precision and navigation success. Increasing to two strides substantially improves performance, suggesting that multiple strides provide richer supervision signals and enhance the model's ability to capture multi-step dependencies. While three strides further improves trajectory diversity, the gain over two strides is marginal, and in some cases introduces additional noise due to redundant or low-quality sub-trajectories.

Overall, the navigation experiments demonstrate that GUI-ReWalk markedly enhances multi-step decision-making compared with vision–language models of similar scale. On AndroidControl, it shows stronger competence in both fine-grained action execution and higher-level task abstraction, indicating a better balance between low-level precision and high-level planning. On GUI-Odyssey, GUI-ReWalk exhibits greater robustness in structured reasoning and long-horizon action sequencing, suggesting improved generalization to complex decision chains. Relative to other 7B-scale baselines such as Qwen2.5-VL-7B, GUI-ReWalk consistently achieves more reliable performance by leveraging systematic trajectory generation and task-aware supervision. These results highlight its effectiveness as a scalable navigation framework, capable of supporting complex hierarchical planning and robust action decomposition within diverse GUI environments.

## 5 CONCLUSION

In this work, we introduced **GUI-ReWalk**, a reasoning-enhanced framework for synthesizing realistic and diverse GUI interaction trajectories. By unifying stochastic exploration with goal-directed reasoning, GUI-ReWalk captures both the long-tail variability and the structured intent of human-computer interactions. Its multi-stride design enables the construction of long-horizon workflows spanning multiple applications, offering a closer reflection of real-world usage patterns than prior datasets. Extensive evaluations show that training models on GUI-ReWalk yields broader interaction coverage, higher trajectory entropy, and more faithful representations of user intent across diverse benchmarks. Beyond providing a scalable data generation pipeline, GUI-ReWalk underscores the importance of reflective reasoning, error recovery, and platform diversity in advancing GUI agent research, paving the way toward next-generation agents that are both resilient and capable of real-world automation at scale.

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

# A  DATA STATISTICS

## A.1  UNIFIED ACTION SPACE

To ensure consistency and comparability across diverse environments, GUI-ReWalk adopts a Unified Action Space that provides a standardized abstraction of user interactions. As shown in Table 4, this unified space covers both mobile and desktop specific actions while maintaining a shared core set. Following the design in UI-Tras (Qin et al., 2025), we further refine the original Finished() action into two distinct outcomes: Completed() and Infeasible(), enabling agents to distinguish between successful completion and infeasible goals—both critical signals for robust policy learning. Moreover, by reporting action distributions separately for mobile and desktop environments, GUI-ReWalk highlights platform-specific interaction patterns (e.g., scrolling and app-launching on mobile vs. richer mouse/keyboard operations on desktop), providing deeper insight into data characteristics.

## A.2  APPLICATION DIVERSITY

In addition to action-level statistics, we analyze the diversity of applications involved in task execution. As illustrated in Figure 3, tasks span a wide range of categories such as communication, productivity, multimedia, system functions, and browsing, ensuring that trajectories reflect realistic multi-domain usage. Importantly, GUI-ReWalk does not impose fixed constraints on the number of applications. Beyond a set of pre-installed apps, the generative process can autonomously guide the installation of new applications when required, enabling data collection to naturally expand into novel domains. This design closely mirrors how users interact with devices in practice, where workflows evolve dynamically across both familiar and newly introduced apps.

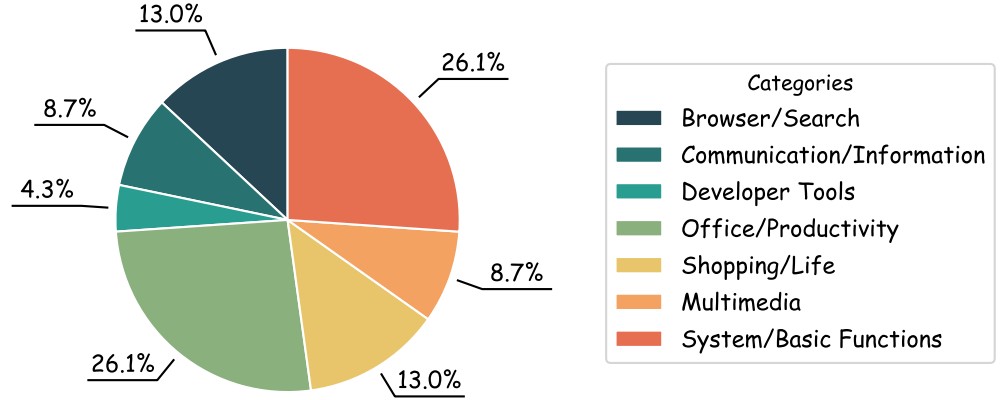

Figure 3: GUI-ReWalk Dataset Composition Across Application Domains.

## A.3  DATASET SCALE AND COMPARISON

Unlike prior datasets that are limited to a single platform or rely solely on human demonstrations (Table 5), GUI-ReWalk spans both mobile and desktop environments, synthesized via a reasoning-enhanced generative process. This design enables large-scale coverage with 50k+ annotated tasks and an average trajectory length of 22.5 steps, surpassing prior datasets. Moreover, by emphasizing long-horizon trajectories with multi-stride structures, GUI-ReWalk better reflects the complexity of real-world workflows across applications.

Table 5: Comparison of GUI-ReWalk and Other GUI Datasets.

| Dataset | Env. | Ann. | Dom/AxT. | Thoughts | Tasks | Avg.Step |
|---|---|---|---|---|---|---|
| AndroidControl (Li et al., 2024) | Mobile | Human | ✓ | Short | 15283 | 5.5 |
| AMEX (Chai et al., 2025) | Mobile | Human | ✗ | ✗ | 2991 | 11.9 |
| AitW (Rawles et al., 2023) | Mobile | Human | ✓ | ✗ | 2346 | 8.1 |
| AitZ (Zhang et al., 2024) | Mobile | Human | ✗ | Short | 1987 | 6.0 |
| GUI-Odyssey (Lu et al., 2024) | Mobile | Human | ✗ | ✗ | 7735 | 15.3 |
| OS-Genesis (Sun et al., 2025) | Mobile&Web | Model | ✓ | Short | 2451 | 6.4 |
| WonderBread (Wornow et al., 2024) | Web | Human | ✓ | ✗ | 598 | 8.4 |
| AgentTrek (Xu et al., 2025a) | Web | Model | ✓ | Short | 10398 | 12.1 |
| Mind2Web (Deng et al., 2023) | Web | Human | ✓ | ✗ | 2350 | 7.3 |
| GUIAct (Chen et al., 2025c) | Web | Human | ✓ | ✗ | 2482 | 6.7 |
| AgentNet (Chen et al., 2025c) | Desktop | Human | ✓ | Long | 22625 | 18.6 |
| GUI-ReWalk (Ours) | Mobile&Desktop | Model | ✓ | Long | 50k+ | 22.5 |

## B  CASE STUDY

### B.1  CORNER CASE

Due to the inherent stochasticity in both the starting point selection and the intermediate navigation process of our framework, GUI-ReWalk occasionally uncovers rare yet semantically meaningful task trajectories—corner cases that are seldom observed in typical user behavior logs. Such cases are valuable for expanding the model's behavioral coverage and pushing the boundaries of its capability in handling unconventional workflows. One illustrative example occurs within the Settings application, as shown in Figure 4 : starting from the Your Information page, the agent navigates to the

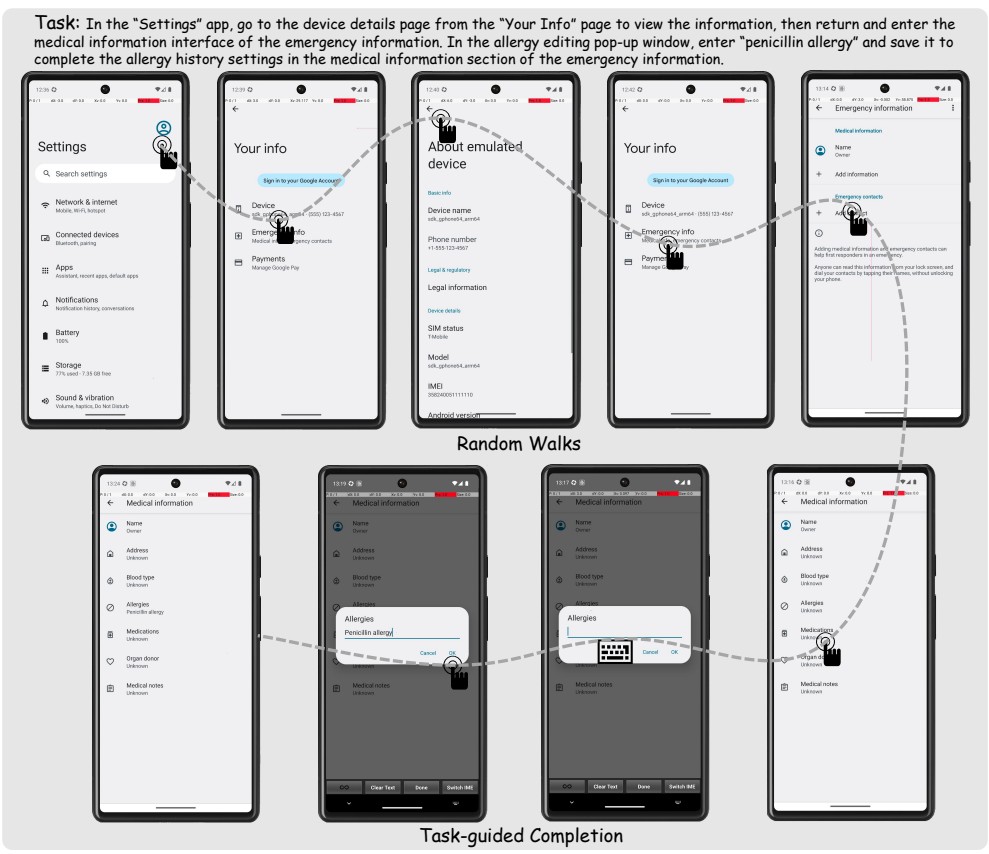

Figure 4: Corner Case Example Demonstrating Rare Yet Coherent GUI Task Trajectories.

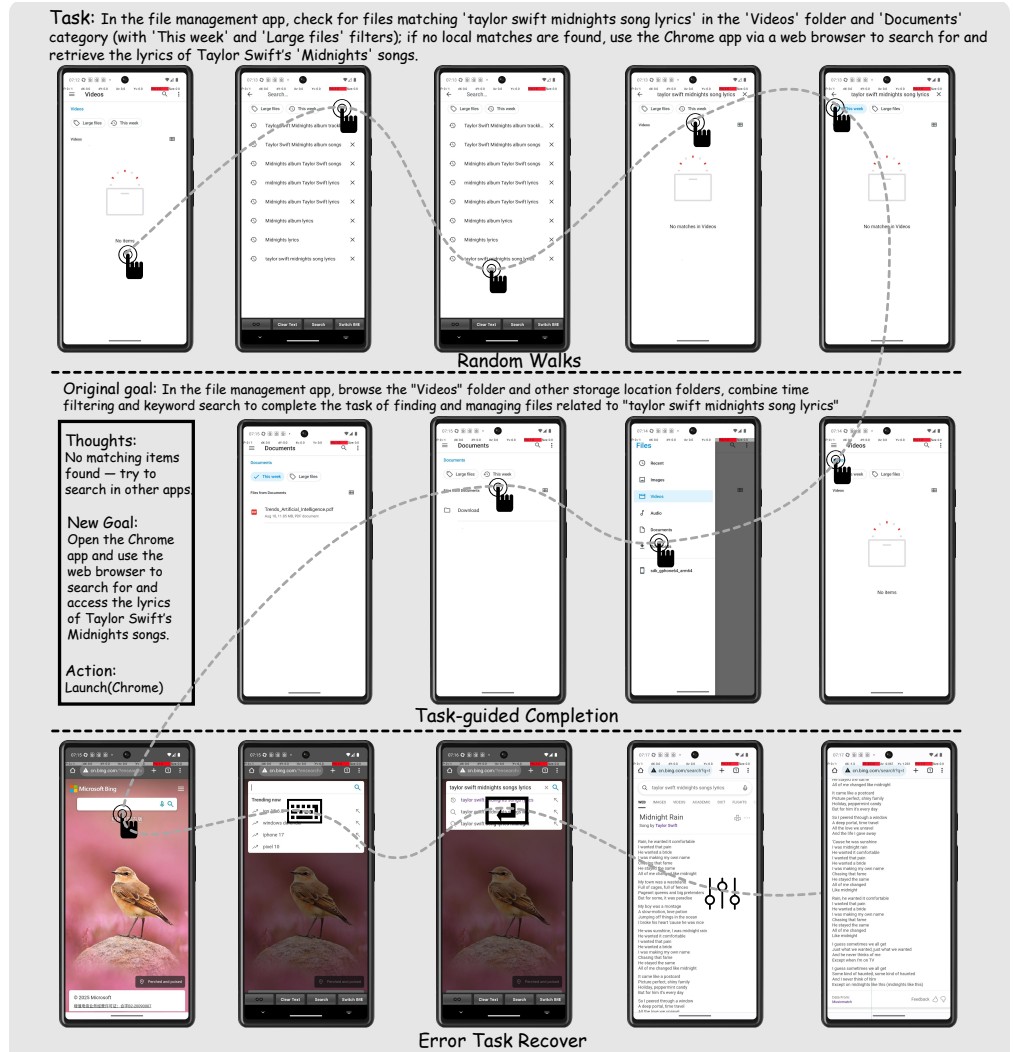

Figure 5: Error Task Recovery Through Reflective Reasoning in GUI-ReWalk

device details page to inspect system information, then returns to the main settings menu before accessing the Emergency Information section. From there, it enters the medical information interface, opens the allergy editing dialog, inputs "Penicillin Allergy", and saves the entry—thus completing the allergy history configuration in the medical information subsection of the emergency settings. This sequence demonstrates the framework's ability to generate coherent, multi-step interactions that traverse atypical paths, thereby revealing functional areas and UI states often underrepresented in standard datasets.

## B.2 ERROR TASK RECOVERY

A unique advantage of GUI-ReWalk lies in its ability to recover from error-prone or infeasible task completions by leveraging reflective reasoning. Since many trajectories are synthesized through task completion and augmentation, the generated goals may occasionally lead to dead ends—either due to infeasible conditions or execution errors. Without intervention, this could trap the agent in repetitive loops or terminal failure states. To address this, GUI-ReWalk equips the reasoning module with the capacity to introspect: upon detecting an infeasible trajectory, the model evaluates whether the failure stems from incorrect execution or from the intrinsic impossibility of the goal. In the latter

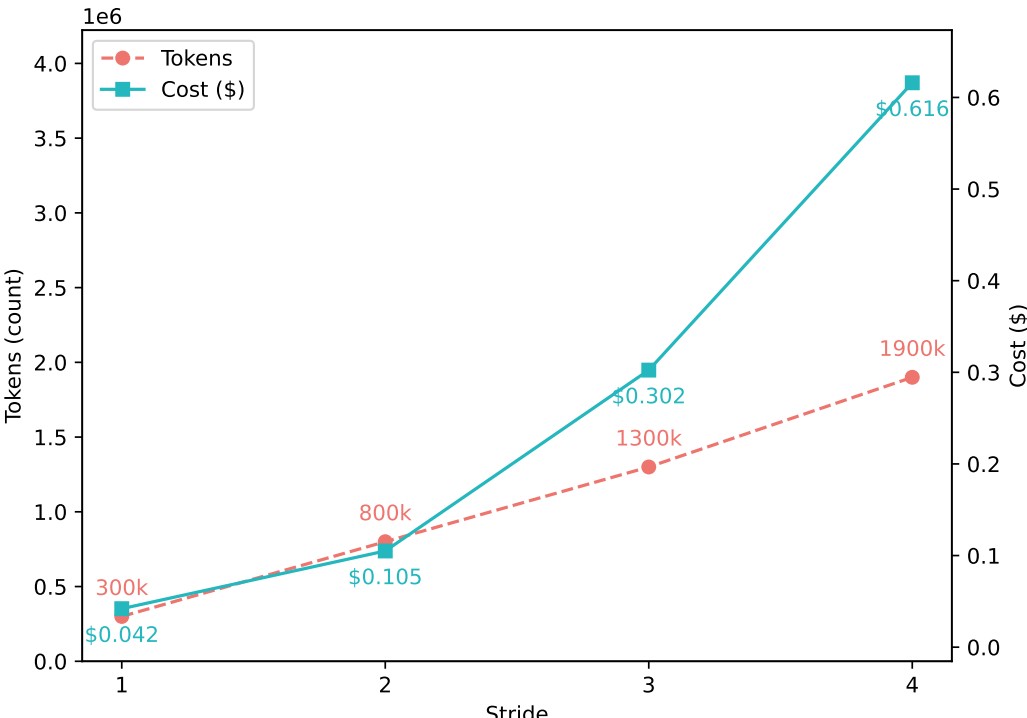

Figure 6: Scaling of Token Usage and Cost with Increasing Strides in GUI-ReWalk.

case, the system revises the original goal into a new, executable objective, thereby restoring progress and ensuring task continuity.

As illustrated in Figure 5, an initial file-search task in a local app proved infeasible, GUI-ReWalk was able to reformulate the goal into a web-based search and successfully complete the objective. Such recoveries enrich the dataset with reflection-driven adaptations, offering agents exposure to trajectories that move from failure to correction—an ability essential for robust and resilient real-world behavior.

## C    DATA COST

To better understand the resource efficiency of GUI-ReWalk, we analyze the token consumption and monetary cost required for generating trajectories of different stride lengths. As shown in Figure 6, the average usable trajectory incurs approximately 300k, 800k, 1300k, and 1900k tokens for 1- to 4-stride tasks, corresponding to average costs of $0.042, $0.105, $0.302, and $0.616, respectively. We observe a near-linear growth in both token usage and cost with increasing strides, as shown in Figure 6. However, the cost curve exhibits a steeper rise, reflecting the higher marginal expense of longer reasoning chains.

## D    LIMITATIONS

While GUI-ReWalk demonstrates strong capability in synthesizing realistic and diverse GUI trajectories, several limitations remain.

**Login-related operations.** A key challenge lies in handling scenarios that involve user authentication. Although we enforce constraints to minimize trajectories requiring login steps, random exploration and downstream task execution can still occasionally lead to login pages, as many applications and websites restrict full functionality to authenticated users. To protect user privacy and avoid exposing sensitive credentials, such trajectories are explicitly filtered out. As a result,

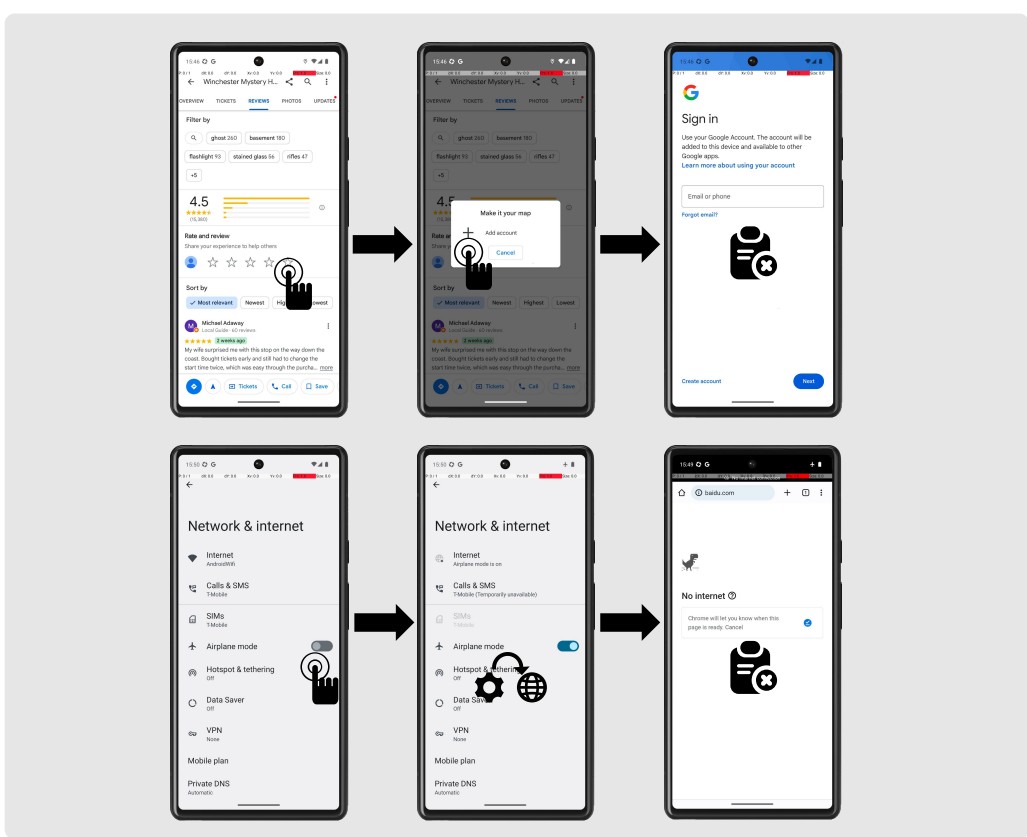

Figure 7: Illustrative Examples of GUI-ReWalk Limitations.

GUI-ReWalk cannot provide coverage for tasks that critically depend on authenticated states, which may limit the completeness of some application workflows. **System-level side effects.** Another

Table 6: Screenspot-Pro results across different domains. Each domain includes Text and Icon grounding.

| Model | CAD | | DEV | | Creative | | Scientific | | Office | | OS | | Avg | | |
|---|---|---|---|---|---|---|---|---|---|---|---|---|---|---|---|
| | Text | Icon | Text | Icon | Text | Icon | Text | Icon | Text | Icon | Text | Icon | Text | Icon | Avg |
| GPT-4o (OpenAI, 2024) | 2.0 | 0.0 | 1.3 | 0.0 | 1.0 | 0.0 | 2.1 | 0.0 | 1.1 | 0.0 | 0.0 | 0.0 | 1.3 | 0.0 | 0.8 |
| SeeClick-9.6B (Cheng et al., 2024) | 2.5 | 0.0 | 0.6 | 0.0 | 1.0 | 0.0 | 3.5 | 0.0 | 1.1 | 0.0 | 2.8 | 0.0 | 1.8 | 0.0 | 1.1 |
| OA-Atlas-7B (Wu et al., 2024) | 12.2 | 4.7 | 33.1 | 1.4 | 28.8 | 2.8 | 37.5 | 7.3 | 33.9 | 5.7 | 27.1 | 4.5 | 28.1 | 4.0 | 18.9 |
| UGround-7B (Gou et al., 2025) | 14.2 | 1.6 | 26.6 | 2.1 | 27.3 | 2.8 | 31.9 | 2.7 | 31.6 | 11.3 | 17.8 | 0.0 | 25.0 | 2.8 | 16.5 |
| UI-TARS-1.5-7B (Qin et al., 2025) | 49.2 | 17.2 | 56.5 | 15.9 | 60.1 | 14.7 | 74.3 | 24.5 | 81.4 | 43.4 | 55.1 | 18.0 | 62.7 | 20.0 | 46.4 |
| Qwen2.5-VL-7B (Bai et al., 2025) | 17.2 | 3.1 | 35.1 | 2.1 | 23.2 | 6.3 | 36.1 | 6.4 | 41.8 | 11.3 | 28.0 | 13.5 | 29.7 | 6.5 | 20.8 |
| **GUI-ReWalk-7B (ours)** | 35.0 | 17.9 | 46.8 | 11.0 | 40.9 | 9.8 | 60.4 | 28.2 | 56.5 | 28.3 | 39.2 | 19.1 | 46.2 | 17.2 | 35.1 |

limitation emerges from system-level operations that inadvertently affect other applications. During random walks or reasoning-guided execution, certain actions in the system settings (e.g., enabling airplane mode, restricting network access for specific apps) can alter global device configurations. Such changes may interrupt network connectivity or disable essential app functionalities, preventing the continuation of subsequent trajectories. As illustrated in Figure 7, these side effects not only reduce usable data but also highlight the inherent complexity of faithfully simulating open-world GUI environments.

Overall, these limitations underline the challenges of balancing privacy preservation, system stability, and data fidelity in large-scale GUI trajectory generation. We consider addressing login-handling mechanisms and isolating system-critical operations as important directions for future work.

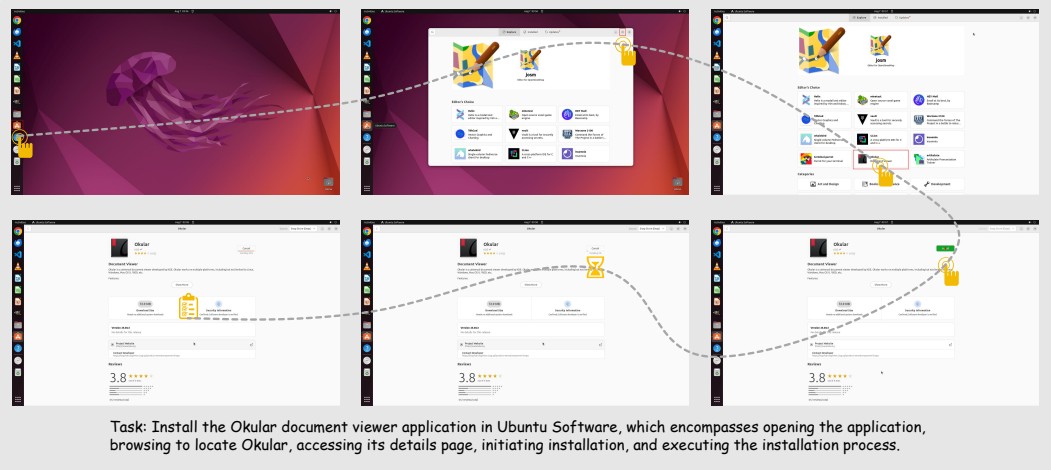

Task: Install the Okular document viewer application in Ubuntu Software, which encompasses opening the application, browsing to locate Okular, accessing its details page, initiating installation, and executing the installation process.

Figure 8: An Example of GUI-ReWalk Trajectory on Desktop.

## D.1 DETAIL RESULT

The detailed results of our evaluation across the three grounding benchmarks are presented in Tables 6 and 7. Specifically, Table 6 reports the sub-task performance on the Screenspot-Pro benchmark, including CAD, DEV, Creative, Scientific, and Office scenarios, each further divided into Text and Icon categories. Table 7 provides fine-grained results on OSWorld-G, covering Text Matching, Element Recognition, Layout Understanding, Fine-grained Manipulation, and Refusal.

Table 7: Results on OS-World-G benchmark. Metrics include Text Matching, Element Recognition, Layout Understanding, Fine-grained Manipulation, and Refusal.

| Model | Text Matching | Element Recognition | Layout Understanding | Fine-grained Manipulation | Refusal | Avg |
|---|---|---|---|---|---|---|
| UGground-7B (Gou et al., 2025) | 51.3 | 40.3 | 43.5 | 24.8 | - | 36.4 |
| UI-TARS-1.5-7B (Qin et al., 2025) | 59.8 | 43.0 | 50.6 | 37.6 | - | 47.5 |
| Qwen2.5-VL-7B (Bai et al., 2025) | 23.0 | 15.5 | 19.0 | 11.4 | - | 16.8 |
| **GUI-ReWalk-7B (ours)** | 35.2 | 30.0 | 31.2 | 16.1 | - | 27.5 |

## E  PROMPTS

### E.1  TASK-GUIDED COMPLETION

---

**SYS NEXT TASK PREDICT**

You are an intelligent assistant observing a user who has just completed a task on their Android mobile or desktop device. Based on this previous task and its context, infer the most likely next task the user would perform. Your goal is to propose a plausible, purposeful, and clearly defined follow-up task that logically continues from the completed one.

Task Generation Requirements:

1. **Logical Continuation**

- The next task must logically build upon the previous one. It should extend or deepen the prior behavior based on user interest, app state, or content.

- Do not repeat, paraphrase, or contradict the previous task.

---

2. **Goal-Oriented and Specific**

- The task must have a clear purpose and a well-defined end state. - Avoid vague descriptions such as "browse more", "explore related content", or "look around".

- Use concrete references (e.g., video titles, place names, keywords, objects, timestamps).

3. **Result-Completeness and Closure**

- The task must include the **final user interaction needed to achieve the goal**, not just the initiation of a process.

- Do **not** stop at intermediate steps like opening an app or search results. - Always include the next logical interaction — such as watching a specific video, opening a particular article, or confirming a key detail — that completes the task.

4. **Completable Within 3 Atomic Actions**

- The task should be feasible with no more than 3 user interactions (e.g., tap, type, select).

- Tasks that require login, account switching, or permission setting are **not allowed**.

5. **Realistic and Executable**

- The task must reflect real usage patterns and be executable in a typical mobile environment.

- Avoid speculative, unsupported, or abstract behaviors.

6. **Content-Aware** - Leverage the context of the prior task: topic, keywords, apps used, content viewed, and user intent.

7. **No Communication Tasks** - Do not include actions involving messaging, emailing, posting to social media, or sharing content.

Output Instructions:

Respond in the following JSON format:

{

"thoughts": "<Detailed reasoning: Why this next task logically follows? How it continues user intent? Why it reaches a meaningful goal within constraints?>",

"task": "<Concrete, result-driven, executable next task with a clear end state>",

"action": "<The first UI action the user would take to begin this task>",

"app": "<The app used to perform this task>"

}

## E.2 CROSS-APPLICATION TASK INITIATION

### SYS CROSS APP NEXT TASK PREDICT

You are an intelligent assistant observing a user who just completed a task on their Android mobile or desktop device. The user is now about to switch apps to perform the next most likely task. Your goal is to propose a plausible, goal-oriented, and clearly defined next task that logically follows from the previous one — but must be completed in a different app, chosen from the list below:

['chrome', 'Map', ...... 'Settings', 'Clock', 'Message']

Task Generation Requirements:

1. **Cross-App Transition**

- The task must take place in a different app from the one just used. - The new app must be selected from the provided list.

- Do not continue in or return to the current app.

2. **Logical Continuation**

- The task must logically extend the user's prior goal, intent, or content.

- Use topic, keywords, content type, or interest signals from the prior task to justify the transition.

3. **Result-Completeness and Closure** - The task must reach a clearly **observable outcome** (e.g., opening and watching a specific video, reading an article, confirming a location).

- Do **not** stop at intermediate actions like opening the app, reaching a search page, or listing results.

- Always include the follow-up interaction that completes the intended action.

4. **Clarity and Specificity** - Avoid vague terms like "explore", "browse", "check out more".

- Use real or plausible entities: keywords, names, places, or identifiers.

5. **Minimal Interaction Constraint** - The entire task must be achievable within 2 atomic actions (e.g., tap + type, tap + select).

6. **Feasibility** - Do not propose tasks requiring login, sharing, permission granting, or complex navigation.

- The task must be executable in a standard Android or desktop environment.

Output Instructions:

Respond in the following JSON format:

{

"thoughts": "<Explain why this app is chosen and why the task is a logical continuation of the previous one. Justify that it is feasible, relevant, and result-complete.>",

"task": "<Specific, result-oriented next task completed in a different app>",

"action": "<First action the user would take to begin this task>",

"app": "<The app name chosen from the list where the task will be completed>"

}

---

### USER TASK PREDICT PROMPT

Given the history {task-history}, what would be a followup task?

---

### E.3 RETROSPECTIVE ANNOTATION

### SYS TASK SUMMARY

You are given a complete sequence of user actions performed on a computer or mobile device. For each step, you have access to:

- The corresponding screenshot,

- An inferred high-level instruction (describing the likely intent of the user at that step),

- A summarized subtask description derived from groups of related actions.

Your goal is to summarize the entire user session as a **single, complete, and clearly defined task** that was accomplished by performing these actions in sequence. This task should reflect the actual goal the user achieved — not just transient interactions, UI distractions, or speculative behavior.

Summary Requirements:

1. **Task-Oriented Abstraction**

- Focus on summarizing the **goal-directed behavior** completed across the session.

- Do **not** include irrelevant, passive, or system-generated steps (e.g., default text suggestions, placeholder content, momentary misclicks).

- Only describe actions that clearly contributed to the user's intent.

2. **Completeness**

- Cover the full behavioral trace, including the final meaningful step.

- Avoid premature truncation or skipping the ending goal.

3. **Relevance Filtering**

- Exclude intermediate or background steps that do not meaningfully advance the user's task (e.g., UI defaults, empty search suggestions).

- Ignore content not clearly chosen or interacted with by the user.

4. **Clarity and Specificity**

- Use precise language to describe what was done and why.

- For search, clearly state the keyword or target topic.

- Avoid vague or generic phrases such as "browse content", "explore topics", or "view related info".

5. **Logical Coherence**

- Ensure the steps form a **cohesive and purposeful progression**, not a fragmented list.

- If multiple apps are used, explain how they connect toward the same goal.

Output Style:

- Write the task in a **formal, instructional tone**, as if specifying a goal in a product spec or user intent model. - Avoid uncertain or hypothetical phrasing (e.g., "might have", "possibly", "if needed").

- The final output should be **specific, executable, and self-contained**.

Output Format:

{

"thoughts": "<Detailed reasoning and interpretation of the user's session, focusing on core goal, meaningful steps, and logical structure. Discard irrelevant or passive actions.>",

"task": "<Final task description, formal and precise, covering only essential, purposeful actions>"

}

---

**USER TASK PREDICT PROMPT**

Given the set of screenshots of actions, instructions {instruction}, and summary histories {summary-list} what would be a single task description that will be accomplished by performing these actions in the given sequence?

---

### E.4 TASK RECOVERY

---

**SYS RECOVERED TASK PREDICT**

You are an intelligent assistant helping to recover from a failed or stuck mobile/desktop automation task.

You will be given:

- The user's **original goal**

- A **summary** of attempted actions and why they failed

- The **current screen description** (visible app and UI state)

Your job:

Reformulate the task so it is **actually achievable**, while preserving the user's **core intent** and maintaining logical continuity between tasks.

—

Recovery Decision Process

1. **Feasibility Assessment**

- Based on the 'summary' and 'current screen', determine if the original goal is realistically achievable in the current environment.

- Criteria for "Not Achievable":

– The target object/content does not exist or cannot be found

– The app lacks the required function or permission

– The path has been fully tried with no results

2. **If Achievable → Path Adjustment Mode**

- Keep the same overall intent but **change the execution path** (use different UI elements, menus, search terms, or filters).

- Explicitly avoid any UI element, keyword, or path already used in failed attempts.

3. **If NOT Achievable → Intent Reconstruction Mode**

- Keep the **main topic keywords** (e.g., subject name, file name, product title).

- Change the environment, app, or method to achieve a **related but feasible outcome**.

- Examples:

– If searching for a file failed → switch to opening a website or app to download it

– If opening a folder failed → use an alternative source for similar content

- The new goal can differ significantly from the original in method, but must stay relevant to the original intent.

4. **Goal Requirements**

- Must have a concrete end state achievable within 3 atomic actions.

---

- Avoid vague "explore more" or "browse around" type tasks.

- No login, messaging, posting, or speculative actions without visible context.

5. **Reasoning Requirements**

- In 'thoughts', explicitly state: - Feasibility judgment (Achievable / Not Achievable)

- Failure reason from summary

- Which mode was chosen (Path Adjustment / Intent Reconstruction)

- How the new goal differs in execution but keeps logical continuity

—

- If the attempted actions repeatedly fail due to the target object being non-existent or non-interactive, do NOT rephrase or retry the same goal.

- Instead, switch to a new but logically related goal by:

1. Retaining the core topic keywords .

2. Redirecting the user to an alternative but feasible outcome .

- This ensures the task moves forward instead of being trapped in repeated reformulations.

Output Format

Respond in the following JSON format:

{

"thoughts": "<Feasibility check, mode chosen, banned paths, reasoning for changes, and why success is more likely>",

"task": "<Revised, achievable, goal-driven task with a clear end state>", "action": "<First UI action to begin this task>",

"app": "<The Android app to perform this task>"

}

---

**USER RECOVERED TASK PREDICT**

Given the action summary {summary} and original goal {goal}, what would be a followup task?"

## F   ETHICS STATEMENT

All authors of this submission have read and agree to adhere to the ICLR Code of Ethics (https://iclr.cc/public/CodeOfEthics). We confirm that our research complies with all applicable ethical guidelines, including those related to research integrity, data handling, and potential conflicts of interest. Our work does not involve human subjects, sensitive data, or applications with potential harmful impacts. To ensure transparency, we have disclosed all funding sources and affiliations in the main text. Any potential ethical concerns, such as fairness or bias in our proposed methodology, have been carefully considered and addressed in the main paper.

## G   THE USE OF LLM

In this work, we adhered to ethical guidelines regarding the use of large language models (LLMs) in academic writing. Specifically, LLMs were employed solely for polishing the manuscript after the initial drafting was completed by the authors. This involved minor refinements such as improving sentence structure, enhancing clarity, correcting grammatical errors, and ensuring consistent terminology, without generating any original content, ideas, methodologies, or results. All core contri-

butions, including the research design, experiments, analysis, and conclusions, were conceived and executed independently by the human authors. This limited application aligns with ICLR's policies on AI-assisted writing to maintain the integrity and originality of the submission.

# H REPRODUCIBILITY STATEMENT

To ensure reproducibility, we have provided comprehensive details to enable replication of our results. Upon acceptance of the paper, we will immediately open-source all code, datasets, and model checkpoints, which will be made publicly available at a specified repository (to be provided in the camera-ready version). All experimental details, including hyperparameters, setups, and evaluation metrics, are clearly described in the Experiments section of the main paper. Additionally, all prompts used in our experiments are fully detailed in Appendix. These resources collectively ensure that our experiments can be independently verified and reproduced.

