# OpenReview forum: "GUI-ReWalk: Massive Data Generation for GUI Agent via Stochastic Exploration and Intent-Aware Reasoning"
_ICLR.cc/2026/Conference — ICLR 2026 Conference Withdrawn Submission_

### Official Review · Reviewer_NRny · 2025-10-23

**Soundness:** 3
**Presentation:** 3
**Contribution:** 3
**Rating:** 2
**Confidence:** 3

**Summary:**

GUI-ReWalk is a framework for generating large-scale, realistic GUI interaction data to train GUI agents. It combines two stages: a stochastic exploration phase that mimics human trial-and-error, followed by an intent-aware reasoning phase where an LLM infers goals and guides purposeful actions. It also supports multi-step workflows across different applications. The authors use this framework to create a dataset and train GUI-ReWalk-7B, which achieves better interaction coverage, higher diversity, and stronger grounding/navigation performance on benchmarks like Screenspot-Pro, OSWorld-G, AndroidControl, and GUI-Odyssey.

**Strengths:**

1. GUI-ReWalk has a clear, human-like data generation design. The paper unifies a random-walk “explore” phase with an intent-aware, reasoning-guided “act” phase, so trajectories are both diverse and purposeful. It also supports multi-stride, cross-application workflows, mirroring how users chain tasks across apps.

2. GUI interactions are framed as a hierarchical MDP with strides and goal inference, while the exploration phase is modeled as a Markov chain; this gives a crisp structure for training agents to switch from probing to goal-directed behavior. The task-guided phase uses LLM policies conditioned on inferred goals, capturing purposeful action planning.

3. Retrospective annotation turns raw trajectories into step-level instructions and high-level tasks without human labeling. A built-in task-recovery mechanism revises goals when progress stalls, improving reliability in messy real interfaces.

4. Training GUI-ReWalk-7B yields notable gains on grounding and better navigation, including higher success rates and accuracy on AndroidControl and GUI-Odyssey. The breadth of evaluations strengthens the claim that the data generation pipeline transfers across settings.

5. The framework targets multi-platform coverage, long-tail patterns, and long-horizon workflows, and reports higher trajectory entropy and more faithful user intent (with human checks). This focus on realism improves the chances that trained agents generalize beyond narrow, scripted tasks.

**Weaknesses:**

1. Gains in Table 2 are modest and not statistically significant compared to the base model Qwen2.5-VL-7B, occasionally even degrading performance. Baselines like OS-Genesis-7B are finetuned from Qwen2-VL-7B, while GUI-ReWalk-7B is finetuned from Qwen2.5-VL-7B. This makes for an uneven comparison, yet the performance is still not better.

2. Human evaluation lacks detail in the main text. You state that results are validated by human evaluations, but the protocol, annotator pool, prompts, and inter-rater agreement are not specified here. Include a short methods box with rater instructions, sampling, agreement stats, and example rubrics to make the claim clear.

3. Quality control for automated annotation is under-explained. Retrospective annotation fully automates step and task labels, but failure modes (hallucinated steps, incorrect goal summaries) are not analyzed.

4. Data realism vs. randomness needs tighter filtering. The random-walk phase can generate actions that humans rarely do, which may inflate entropy but hurt policy learning. Consider adding filters (e.g., UI affordance checks, heuristics) and reporting how often random steps are pruned, along with a study linking entropy to downstream accuracy.

5. Improvements are uneven across benchmarks and deserve deeper analysis. Gains on UI-Vision are modest, and the paper would benefit from per-category breakdowns and error taxonomies. Add ablations by domain and failure-case studies to show where the method helps most and where it still struggles.

6. The stride ablation shows non-monotonic trade-offs and marginal gains beyond two strides, hinting at noise or redundancy. Extend the analysis with longer horizons, curriculum schedules, and data-budget sweeps.

**Questions:**

1. How do you prevent unrealistic random-walk behavior and ensure the generated action mix matches real users and useful behavior, and can you share statistics on the proportion of filtered steps or action types across platforms?

---

### Official Review · Reviewer_evZf · 2025-10-30

**Soundness:** 2
**Presentation:** 2
**Contribution:** 2
**Rating:** 4
**Confidence:** 4

**Summary:**

This paper proposes GUI-ReWalk, a hierarchical framework for generating large-scale synthetic data to train and evaluate GUI agents. GUI-ReWalk mimics human interaction through staged stochastic exploration (random walk) and intent-aware reasoning (goal-driven actions), enabling multi-stride, cross-application GUI task generation. The framework includes retrospective annotation using LLMs and a task recovery scheme to ensure robust, coherent, and semantically rich data. The resulting synthetic dataset is used to train GUI-ReWalk-7B, which is evaluated against multiple grounding and navigation benchmarks—such as Screenspot-Pro, OSWorld-G, AndroidControl, and GUI-Odyssey—showing improved trajectory diversity, intent alignment, and overall task success compared to strong baselines.

**Strengths:**

1. GUI-ReWalk effectively models both the randomness of human GUI exploration and the goal-driven nature of digital workflows, leading to datasets that present greater diversity (long-tail behaviors), richer structure, and intentionality than prior purely random or solely scripted synthesis pipelines.

2. The framework supports long-horizon workflows traversing multiple apps, a feature many existing datasets lack. Figure 1 visually highlights these novel axes—multi-environment, long-tail, reflection, multi-stride—that position GUI-ReWalk toward real-world applicability, while Figure 2 clarifies the phased architecture for generation and annotation.

**Weaknesses:**

1. The related works section is thorough on established datasets and agent models but omits discussion of recent, closely aligned 2025 works introducing advanced GUI data generation and agent frameworks (for example, GUI-Xplore, ShowUI, Magentic-UI, WebGuard, and Cognitive Kernel-Pro). Neglecting these highly relevant works undermines the positioning and fails to clarify the case for scientific advancement over these alternatives.

2. While the method leans heavily on LLMs for goal inference, annotation, and task recovery, there is limited empirical or theoretical analysis on prompt sensitivity, LLM failures, or hallucination-driven errors. Section E of the appendix lists prompts, but there is no systematic study of prompt robustness or ablation—raising concerns on reproducibility and dataset artifacts caused by LLM-specific behaviors.

3. The pipeline mentions LLM-filtered SFT data for navigation and post-hoc removal of privacy or system-unstable episodes (Appendix D), but there is no quantification of dropped, noisy, or failed samples, nor discussions on dataset bias or class imbalance. Figure 6 gives a cost analysis, but not an audit trail for data quality impact.

4. The mathematical modeling formalism (Section 3) is largely an adaptation of standard MDP/hMDP representations and reward structures (see, e.g., Andrychowicz et al. 2017; Kaelbling et al. 1998), rather than an innovation. There is little theoretical development beyond this translation to the GUI domain. If the framework is to be championed on scientific advance rather than engineering scalability, more rigorous theoretical or statistical grounding is expected.

5. The paper reports impressive absolute and relative gains across tasks in Tables 1 and 2, but omits standard deviations, confidence intervals, and n-values for repeated runs. This calls into question the statistical significance of these reported improvements and hinders robust external evaluation.

**Questions:**

1. Can the authors provide empirical ablations on LLM prompt sensitivity and downstream impact of prompt variation in task annotation, recovery, and goal inference?
2. Are there quantitative metrics on the proportion of trajectories filtered/removed due to privacy conflicts, infeasible system actions, or failed LLM annotation? What is the distribution of dropped samples and how might this affect real-world agent generalization?

3. Please justify, in concrete terms, the superiority claims over contemporaneous works (e.g., GUI-Xplore, ShowUI) on either coverage, quality, or cost—ideally with empirical results or at least thorough qualitative/analytical comparison.

4. What are the practical implications of excluding all login/auth flows from your dataset in terms of coverage for real digital workflows, especially in workflow-heavy applications?

5. Do you have mechanisms or analyses for detecting and correcting dataset bias—e.g., overrepresentation of trivial or redundant workflows, or severe action/domain class imbalance?

---

### Official Review · Reviewer_1BJC · 2025-11-01

**Soundness:** 2
**Presentation:** 3
**Contribution:** 2
**Rating:** 4
**Confidence:** 4

**Summary:**

This paper proposes **GUI-ReWalk**, a multi-stage framework for generating large-scale, realistic GUI interaction trajectories. It combines **stochastic exploration** that mimics human trial-and-error with **intent-aware reasoning** for goal-directed actions, formalized as a **hierarchical Markov Decision Process**.

The framework includes random exploration, task-guided completion, cross-application task initiation, retrospective annotation, and task recovery. A 7B model, **GUI-ReWalk-7B**, built on Qwen2.5-VL-7B, is trained on the generated data and achieves notable improvements on multiple GUI benchmarks (e.g., Screenspot-Pro, OSWorld-G, GUI-Odyssey).

The main contributions are:

1. A scalable data generation pipeline unifying exploration and reasoning;
2. A hierarchical formulation capturing human-like GUI behavior;
3. Strong empirical results demonstrating improved grounding and navigation performance.

**Strengths:**

The paper’s main strength lies in the **scale and temporal depth** of its generated dataset. The proposed GUI-ReWalk pipeline produces roughly **50k GUI trajectories** across **multiple environments (mobile, desktop, and cross-application)**—a clear increase over prior synthetic datasets that typically contain fewer than 20k samples. Moreover, these trajectories are **long-horizon**, averaging **20 or more steps** compared to the ~10 steps common in earlier works, which provides richer supervision for reasoning and planning. While the methodological novelty is modest, the framework makes a **practical contribution** by demonstrating a scalable and automated way to synthesize diverse, long-horizon GUI interaction data for benchmarking and model training.

**Weaknesses:**

* There is a citation inconsistency in the dataset comparison table — AgentNet (Chen et al., 2025c) appears to cite GUI-Course rather than the correct source, which raises concerns about the accuracy of reported baselines and data attribution.
* The experimental evaluation is not solid or convincing relative to the paper’s stated goals. The core objective of GUI agents is end-to-end task completion in real, interactive environments, yet the experiments mainly focus on static GUI grounding and single-platform mobile navigation benchmarks. Although the authors claim to have generated a large-scale dataset covering multiple environments (desktop, mobile, and cross-application), the evaluations fail to reflect this diversity.
* In particular, the reported gains on AndroidControl are marginal, and the work does not include results on realistic interactive benchmarks such as AndroidWorld or OSWorld, which are more representative of real-world GUI agent performance. As a result, it is unclear whether the proposed data generation pipeline genuinely improves the agent’s capability to perform long-horizon, multi-platform tasks beyond synthetic or static test settings.

**Questions:**

- Desktop coverage — You claim multi-platform data (mobile/desktop/cross-app), but experiments center on grounding and a single-platform mobile navigation task. Please add desktop results—at least a static desktop benchmark (e.g., AgentNetBench)—to substantiate the multi-platform claim.
- End-to-end effectiveness — The core goal is E2E task completion in real, interactive environments. Current results emphasize static grounding and AndroidControl, where gains are modest. Please report on interactive benchmarks like OSWorld (interactive) and AndroidWorld (e.g., SR/long-horizon metrics) to demonstrate that your dataset actually improves end-to-end performance.

---

### Note · Authors · 2026-01-23

**Comment:**

We sincerely appreciate the time and effort the reviewers dedicated to evaluating our work. We have carefully considered the constructive feedback provided, particularly the suggestions regarding the need for comprehensive evaluation in end-to-end interactive environments (e.g., OSWorld) and more extensive comparisons with the latest related works.

After a thorough assessment, we recognize that addressing these concerns rigorously requires conducting a series of new, large-scale experiments and substantial revisions to the manuscript. We have determined that these essential improvements cannot be adequately completed and validated within the constraints of the rebuttal period. To uphold the standards of academic integrity and ensure the reliability of our experimental results, we have decided to withdraw the paper at this stage.

We intend to dedicate the necessary time to conduct these supplementary experiments and incorporate the reviewers' insights to strengthen the work for a future submission.

Thank you again for your valuable feedback and service to the community.

Sincerely,

The Authors

**Withdrawal Confirmation:**

I have read and agree with the venue's withdrawal policy on behalf of myself and my co-authors.